# Crocodile Oil Modulates Inflammation and Immune Responses in LPS-Stimulated RAW 264.7 Macrophages

**DOI:** 10.3390/molecules27123784

**Published:** 2022-06-12

**Authors:** Metas Ngernjan, Atcharaporn Ontawong, Narissara Lailerd, Kriangsak Mengamphan, Sureeporn Sarapirom, Doungporn Amornlerdpison

**Affiliations:** 1Faculty of Fisheries Technology and Aquatic Resources, Maejo University, Chiang Mai 50290, Thailand; themaetas@gmail.com; 2Division of Physiology, School of Medical Sciences, University of Phayao, Phayao 56000, Thailand; atcharaporn.on@up.ac.th; 3Department of Physiology, Faculty of Medicine, Chiang Mai University, Chiang Mai 50200, Thailand; narissara.lailerd@cmu.ac.th; 4Center of Excellence in Agricultural Innovation for Graduate Entrepreneur, Maejo University, Chiang Mai 50290, Thailand; kriang1122sak@gmail.com; 5Applied Physics, Faculty of Science, Maejo University, Chiang Mai 50290, Thailand; ssarapirom@gmail.com

**Keywords:** anti-inflammatory, crocodile oil, DNA damage, immune response, inflammatory cytokine

## Abstract

Crocodile oil (CO) is generated from the fatty tissues of crocodiles as a by-product of commercial aquaculture. CO is extensively applied in the treatment of illnesses including asthma, emphysema, skin ulcers, and cancer, as well as wound healing. Whether CO has anti-inflammatory properties and encourages an immune response remains uncertain. The impact of CO on inflammatory conditions in lipopolysaccharide (LPS)-stimulated RAW 264.7 cells and the mechanisms behind it were examined in this work. Cells were treated with 0.125–2% CO dissolved in 0.5% propylene glycol with or without LPS. The production and expression of inflammatory cytokines and mediators were also examined in this research. CO reduced the synthesis and gene expression of interleukin-6 (IL-6). Consistently, CO inhibited the expression and synthesis of inflammatory markers including cyclooxygenase-2 (COX-2), prostaglandin E2 (PGE2), nitric oxide (NO), and nuclear factor kappa B (NF-κB). Furthermore, CO reduced the effects of DNA damage. CO also increased the cell-cycle regulators, cyclins D2 and E2, which improved the immunological response. CO might thus be produced as a nutraceutical supplement to help avoid inflammatory diseases.

## 1. Introduction

The freshwater crocodile, *Crocodylus siamensis,* is found in Southeast Asia and it is the world’s most endangered crocodilian species. Thailand’s Department of Fisheries has promoted crocodile breeding to increase exports while encouraging conservation. Commercial crocodile breeding occurs in Thailand, where at least one million reptiles are raised. Heavy metals, radiation, and pathogenic microorganisms are no match for a crocodile’s resistance to these threats to its health. Earlier studies have shown that the lysates of several organs from the crocodile’s body (heart, lungs, liver, brain intestines, bile, and gallbladder) have antibacterial anti-tumor effects [1], with antibacterial and wound-healing qualities features of crocodile blood [2]. Crocodile oil (CO) was shown to be effective in the treatment of asthma, emphysema, flu, persistent cough, phlegm-producing mucus, skin ulcers, burns, cancer, and wounds [3,4,5]. Moreover, in traditional remedies, such as traditional Chinese and Southeast Asian medicine, CO and its derivatives are used as ointments for treating burns and scalds [6,7]. Other cosmetic goods made from CO include oils, pain relievers (such as pain-relief balm), and skin lotions. Our previous studies found that omega-9 fatty acid rich in fish oil exerts antidiabetic and anti-inflammatory effects in type 2 diabetic rats and RAW 264.7 models, respectively [8,9]. Despite this, the mechanisms through which CO has anti-inflammatory and immunomodulatory effects are still a mystery.

Inflammation is caused by pathogens, toxins, chemicals, and radiation. Chronic inflammation leads to an increased risk for diabetes, heart disease, cancer, and rheumatoid arthritis. Biologically, the immune system tries to fight against foreign bodies and dangerous chemicals. Immune regulation is also linked to cell-cycle regulators. The gene and protein for the tumor protein p53 (p53) are critical in the fight against viral illnesses [10]. The immune-mediated inflammation seen in p27−/− mice is also linked to the cyclin-dependent kinase inhibitor p27 (p27) [11]. Additionally, a healthy immune system and homeostasis depend on the cell-cycle regulators, cyclins D and E. [12]. Toxic stimuli are eliminated and healing processes are initiated when the immune system is stimulated resulting in inflammation [13]. Macrophages produce a wide range of growth factors and cytokines vital for immunomodulation during inflammatory conditions [14].

A wide range of cytokines produced by macrophages play a vital role in immunomodulation during inflammation. Interleukin-1β and IL-6 are two cytokines that play a key role in the inflammatory condition. Multiple sclerosis, rheumatoid arthritis, Alzheimer’s disease, and inflammatory bowel disease (IBD) are caused by chronic inflammation [15,16]. Consequently, it is vital to devise a plan for reducing inflammation while boosting the immune system. Research on the impacts of CO on LPS-stimulated macrophage cells and the underlying processes was carried out in this work.

## 2. Results

### 2.1. Fatty Acid Composition of CO

There was no odor or color to the CO extracted using microwaves. From the fat tissue, the CO output was 67.50%. There were 28.61% saturated and 71.37% unsaturated fatty acids in CO, and the proportions of omega-3 (1.4%), 6 (23.70%), and 9 (41.07%) are shown in Table 1. Monounsaturated fatty acid (MUFA) accounted for 46.02% of the total. As a result, CO’s low saturated fat content and high omega-9 fatty acid content make it a potentially helpful oil.

### 2.2. CO Inhibits the Generation of Pro-Inflammatory Cytokines in LPS-Activated Cells

To evaluate the anti-inflammatory effect of CO on the production of proinflammatory cytokines, the pro-inflammatory cytokine levels were determined using an ELISA assay. The results show that the IL-6, IL-1β, and TNF-α levels were higher in the LPS group than in the control group (Figure 1A–C), indicating that LPS promotes inflammation. Furthermore, IL-6 production was considerably reduced by using CO at a concentration of 0.25–2% by 1.4–3.6 times compared to the cells stimulated by LPS (Figure 1A). Also, celecoxib (CX) inhibited the production of pro-inflammatory cytokines in the same manner. In terms of the IL-1β and TNF-α levels, however, there were no anti-inhibitory effects from CO. Furthermore, CO (0.25–2%) and CX had no significant influence on RAW 264.7 cell viability compared to the control group (Figure 1D). CO seems to have anti-inflammatory properties without cytotoxicity, based on these findings. As a result, CO with a concentration of 2% was chosen for future investigation.

### 2.3. CO Decreases IL-6 mRNA Expression in LPS-Activated Cells

To study the impact of CO on the inflammatory gene expression, we performed a quantitative PCR. The results show that IL-6, IL-1β, and TNF-α were all more highly expressed in the LPS-treated cells than in the control cells. However, IL-6 gene expression was suppressed in the cells treated with CO (Figure 2). These results suggest that CO inhibits the transcription and translation of pro-inflammatory cytokines in activated macrophages.

### 2.4. CO Reduces Inflammatory Mediators in LPS-Induced Cells

To determine the effects of CO on the production of inflammatory mediators, the mediator levels were determined using Griess, qPCR, and an ELISA assay, respectively. The results show the generation of nitric oxide (NO) (Figure 3A). Cells exposed to LPS for 24 h increased their NO generation compared to the control group. In addition, CO and CX reduced NO levels in LPS-stimulated cells compared to the cells stimulated by LPS alone. Furthermore, CO’s effects on the inflammatory mediators, COX-2 and PGE2, were examined. LPS-treated cells increased COX-2 mRNA expression and produced PGE2. Furthermore, CO significantly reduced the expression of COX-2 mRNA, which in turn lowered the levels of PGE2 (Figure 3B,C). Similarly, CX also decreased PGE2 production and COX-2 expression. According to these findings, CO inhibits COX-2 and PGE2 in LPS-activated macrophage cells, reducing the inflammatory response.

### 2.5. CO suppresses NF-κB p65 Activation in LPS-Induced Cells

The influence of CO on NF-κB p65 activation was determined using fluorescence staining. According to the results presented in Figure 4, LPS-induced inflammation cells elevated the NF-κB p65 activation; however, this activation was prevented in the CO-treated group. Furthermore, CX also reduced NF-κB p65 activity in cells induced with LPS. These findings imply that CO’s anti-inflammatory properties may be attributable to its ability to suppress the NF-κB pathway.

### 2.6. CO Ameliorates DNA Damage in LPS-Activated Cells

Further investigation into the impact of CO on DNA damage was conducted using the 8-OHdG-EIA kit. These LPS-treated cells showed a larger quantity of 8-OHdG than the control cells, as seen in Figure 5. In addition, as was the case in the CX-treated cells, 8-OHdG synthesis was significantly reduced in the CO-treated cells. These findings imply a possible inhibitory effect on the DNA damage response to CO.

### 2.7. CO Triggers an Immunological Response in LPS-Stimulated Cells

To determine the effects of CO on immunological responses, the cell-cycle regulator gene expression was evaluated by qPCR. The results show no significant changes between the control and the CO-treated groups in the mRNA expression of the tumor-suppressor gene (p53) and the cell-cycle inhibitor (p27). However, levels of cyclin D2 and cyclin E2 were more significant in the CO-treated cells, indicating that the cells had been stimulated to enter the cell cycle (Figure 6). According to these findings, CO seems to trigger the immune response by altering the cell-cycle regulator proteins.

## 3. Discussion

The immune system acts as a protective barrier against a wide range of pathogens, bacteria, fungi, viruses, poisons, chemicals, and pharmaceuticals. The innate immune cell, the macrophage, phagocytose and create pro-inflammatory cytokines and mediators in response to pathogens. Several illnesses are caused by an inflammatory state left untreated for an extended period. As a result, decreasing inflammatory cytokines and mediators may be a valuable strategy for disease prevention via immunological modulation. CO’s anti-inflammatory and immunomodulatory properties were examined in this work as well as the mechanisms by which it affects LPS-activated RAW 264.7 cells, as seen in Figure 7.

Omega-9, a monounsaturated fatty acid, is abundant in CO. The anti-proliferative and anti-apoptotic effects of omega-9 on vascular smooth muscle cells are helpful in the development of the atherosclerosis [17]. Aside from these benefits, omega-9 reduces cytokines and enhances neutrophil activity [18]. This research established the anti-inflammatory characteristics of CO high in omega-9, with lower IL-6 production and mRNA expression equivalent to those of nonsteroidal anti-inflammatory medications (NSAIDs). IL-6, IL-1β, and TNF-α transcription and translation were significantly reduced by fish oil rich in omega-9, as shown previously [9]. Pre-treatment with omega-9 oleic acid in septic mice reduced organ damage and boosted survival rates [19] TNF-α, IL-1β, IL-6, IL-12, and cyclooxygenase-2 are all induced by NF-κB, a transcription factor found only in classically activated (M1) macrophages [20]. CO was shown to decrease NF-B activation in the present investigation. According to another recent study, an oil blend from seaweed (low omega-6 to omega-3 and high omega-9) has been shown to promote re-epithelialization and reduce NF-κB levels in a burn model [21]. Additional studies have shown omega-9 oleic acid to have anti-inflammatory properties by inhibiting the LPS-activated nuclear NF-κB and IKK in microglial cells [22]. As a result, the inactivation of the NF-κB pathway contributes to CO’s anti-inflammatory effects.

As well as pro-inflammatory cytokines, activated immune cells emit inflammatory enzymes, for example, iNOS and COX-2 [23]. The iNOS and COX-2 enzymes increase the harmful chemicals including NO and PGE2, respectively. According to this research, CO had similar NSAID-like effects on NO, PGE2, and COX-2, the enzyme responsible for producing PGE2. PGE2 and the enzyme that synthesizes it, COX-2, were shown to be reduced in response to omega-9-rich fish oil [9]. In addition, the LPS-induced iNOS and COX-2 expression and NO and prostaglandin E2 generation were reduced by omega-9 oleic acid in BV2 microglia cells [22]. PPAR, which is involved in inflammation, lipid, glucose metabolism, and other processes, may be activated by omega-9 making it an essential inflammatory mediator [24,25]. A reduction in the inflammatory response, improved survival rates, and inhibition of neutrophil migration have been linked to PPAR-γ activation in inflammation models [26,27]. A PPAR-γ expression-dependent mechanism may underlie omega-9’s anti-inflammatory effects [28].

The inflammatory cells create reactive oxygen and nitrogen species to combat infections that cause DNA damage. CO, similar to NSAIDs, was shown to reduce DNA damage in the present research. According to prior research, in rats with dextran sulfate sodium (DSS) colitis, dietary fish oil reduced colonic inflammation and oxidative DNA damage [29]. According to another study, smokers who took large doses of fish oil had lower 8-OHdG levels in their blood [30]. To maintain the integrity of the immune system, cyclins D2 and E2 play a critical regulatory function in the cell cycle, and CO therapy boosts the expression of these cyclins. *Anemarrhena asphodeloides* extract has been demonstrated to boost the gene expression of cyclins D2 and E2, increasing the immune system’s response [23]. The immunological response was further enhanced by fish oil, which increased the production of cyclins D2 and E2 while suppressing p53 [9].

## 4. Materials and Methods

### 4.1. Chemicals

Lipopolysaccharides (LPS), MTT, and β-nicotinamide adenine dinucleotide phosphate were acquired from Merck (Darmstadt, Germany). BioLegend provided the cytokine ELISA kits (San Diego, CA, USA). Other high-purity chemicals were purchased commercially.

### 4.2. The Process of Making CO

Adipose tissue of freshwater crocodiles (*Crocodylus siamensis*) from Community Enterprise in Thailand was collected. Microwave heating was then used to extract crocodile oil (CO). Microwave heating is a faster process compared to conventional methods that rely on surface heating. In this work, crocodile fat was chopped into small pieces and spread on a sieve tray. The fat from the crocodile had been minced and spread out before being used in this experiment. After 15 min of microwave cooking at 2.54 GHz, 6 kilowatts of electricity were generated from the chopped fat. The created oil had no odor and became transparent in seconds. The CO supernatant determined the oil yield after separating it from the remaining fat. Central Laboratory (Thailand) Co., Ltd., Chiang Mai Branch, a certified lab with international standardization in the field of information technology (ISO17025), tested the fatty acid content of CO using the in-house TE-CH 260 technique of the Association of Official Analytical Chemists 996.06. Briefly, the analysis of the CO fatty acid profile was conducted via gas chromatography (GC) (Agilent technologies, CA, USA, 6890N) equipped with a flame ionization detector (FID) on a capillary column (Supelco, PA, USA, SP-2560: 100 m length, 0.25 mm internal diameter and 0.20 μm film thickness). The column was initially set at 140 °C and held for 5 min, then increased to 250 °C at a rate of 3 °C/minute and held for 17 min at 250 °C. The relative retention time and the individual peak were detected by GC analysis, resulting in a 55 min run time. To quantify the fatty acid concentration, the standard sample peak retention duration and area were compared to the CO sample.

### 4.3. Cell Culture

RAW 264.7 cells were given by the American Type Culture Collection (Manassas, VA, USA). A humidified environment containing 5% CO_2_ was maintained in which cells were seeded in DMEM added with 10% FBS and 1% antibiotic–antimycotic. For subsequent tests, cells were seeded in 6-, 12-, and 96-well plates at a 2.0 × 10^5^ cells/mL density and allowed to develop for three days.

### 4.4. The Vitality of Cells

Cells were tested for vitality using the MTT assay as previously described [31]. The cells were plated at 2.0 × 10^5^ cells/mL in 96-well plates and supplemented with CO at various concentrations (0.125, 0.25, 0.5, 1, and 2%) or celecoxib (CX) at 10 µM for 24 h. Afterward, cells were treated for four hours at 37 °C with the MTT reagent and then added to the DMSO solution. Cell viability was measured using a microplate reader at 570 nm (BioTek, Winooski, VT, USA).

### 4.5. Determination of Inflammatory Markers

The cells were plated in a 12-well plate and supplemented cell with CO at 0.125–2% or CX with or without LPS at 1 μg/mL for 24 h, followed by 2000× *g*, 4 °C, and 10-min centrifugation. Cytokines including IL-6, IL-1β, and TNF-α concentrations were measured at 450 nm using an ELISA kit (BioLegend, San Diego, CA, USA).

### 4.6. Nitric Oxide Level Measurement

The cells were stimulated with LPS at 1 μg/mL and then treated with 2% CO or 10 μM of CX. The cells were subjected to 20 min centrifugation at 4 °C with a speed of 10,000× *g*. A commercial kit was used to measure the level of nitric oxide (NO) in the supernatant (Cayman Chemical, Ann Arbor, MI, USA). The NO concentration was evaluated at 540 nm (BioTek, Winooski, VT, USA).

### 4.7. Measurement of Prostaglandin Level

The cells were stimulated with LPS and then treated with 2% CO or 10 μM CX. Cells were spun at 2000× *g* at 4 °C in a centrifuge for 20 min. The prostaglandin (PGE2) levels were measured using an ELISA kit (Abbkine, Wuhan, China). The PGE2 concentration was assessed using a SynergyTM HT microplate reader with a 450 nm wavelength (BioTek, Winooski, VT, USA).

### 4.8. Determination of NF-κB Activation

Cells exposed to LPS with or without 2% of CO and 10 μM of CX for 24 h were studied. Cells were fixed with 4% paraformaldehyde and permeated with 0.1% Triton X-100. After being treated with primary p65 NF-κB antibody overnight, cells were then incubated for an additional hour with a secondary antibody (Alexa Fluor^®^ 488). The cells were counterstained with Hoechst 33342 for five minutes and the final analysis was performed using a Nikon Eclipse Ni-U fluorescent microscope (Nikon Corporation, Tokyo, Japan).

### 4.9. Evaluation of DNA Damage

The DNA damage marker 8-hydroxy-2′-deoxyguanosine (8-OHdG) was studied using ELISA. One μg/mL LPS with or without CO or CX was added to the cells for 24 h at 37 °C on the day of the experiment. Subsequently, the cells were centrifuged at 10,000× *g* for 20 min at 4 °C. The 8-OHdG concentration was assessed using a SynergyTM HT microplate reader with a 450 nm wavelength (BioTek, Winooski, VT, USA).

### 4.10. Analysis of Inflammatory Gene Using Quantitative Real-Time PCR

Total RNA was extracted and purified from RAW 264.7 cells using TRIzol reagent (Thermo Fisher Scientific, Waltham, MA, USA). First-strand cDNA was obtained using a SensiFAST™ cDNA synthesis kit (Bioline, London, UK). Real-time PCR was performed on QIAquant real-time PCR equipment (QIAGEN, Hilden, Germany). Macrogen (Seoul, Korea) gave the forward and reverse primers which were used at a final concentration of 0.4 mM. TNF-α, IL-1β, IL-6, COX-2, p27, p53, and GAPDH are listed in Table 1 as the primer sets used in qRT-PCR [13,32,33,34,35,36]. The data were expressed in relative fold change (RFC). Each cDNA was amplified using qPCR in five separate experiments.

### 4.11. Analysis of Statistical Data

The data were shown as the mean ± S.E.M. One-way ANOVA and Dunnett’s test examined statistical differences in SPSS version 23. (IBM Corp., Armonk, NY, USA). At a *p*-value of 0.05, differences were deemed significant.

## 5. Conclusions

According to this research, CO contains a high concentration of omega-9 fatty acids and a low concentration of saturated fatty acids. Inflammatory mediators are reduced and DNA damage is improved by reducing the IL-6 production and gene expression and by lowering the inflammatory mediators, NO, COX-2, and PGE2. In addition, the upregulation of the cell-cycle regulators as a result of CO further increases immunomodulation. This means that CO may be effective as a supplement to treat inflammation naturally.

## Figures and Tables

**Figure 1 molecules-27-03784-f001:**
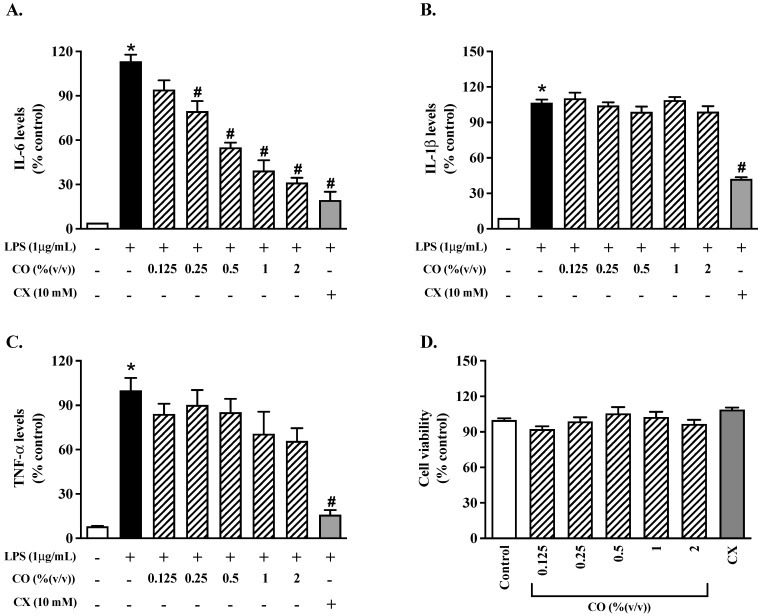
The effects of CO on the production of inflammatory cytokines. For 24 h, cells were treated with varying doses of CO with or without LPS. (**A**–**C**) There were ELISA tests for IL-6, IL-1β, and TNF-α, respectively, to determine the levels of production of these three cytokines. RAW 264.7 cell viability was tested for 24 h at various CO concentrations (**D**). Values displayed are mean ± S.E.M. (*n* = 5), * *p* < 0.05 vs. control and # *p* < 0.05 vs. LPS.

**Figure 2 molecules-27-03784-f002:**
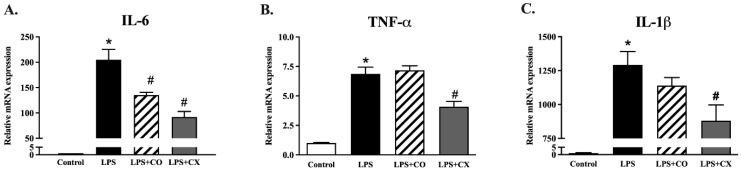
The effects of CO on the gene expression of proinflammatory cytokines. LPS-activated RAW 264.7 cells were incubated with CO for 24 h and the inflammatory cytokine gene expression was measured. During the 24 h of experiments, cells were given either 2% CO or 10 µM CX with or without LPS. qPCR was used for the gene expression of IL-6, IL-1β, and TNF-α (**A**–**C**). Values displayed are mean ± S.E.M (*n* = 5), * *p* < 0.05 vs. control and # *p* < 0.05 vs. LPS.

**Figure 3 molecules-27-03784-f003:**
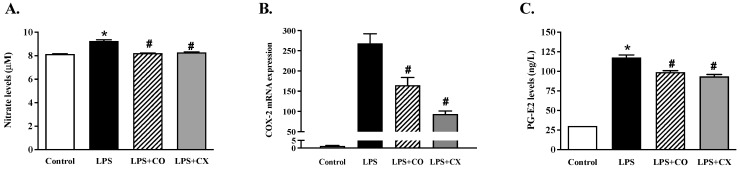
Impact of CO on inflammatory mediators in LPS-activated cells. Cells were activated with LPS and then treated with either 2% CO or 10 µM CX for 24 h. This study evaluated the NO production (**A**), COX-2 mRNA expression (**B**), and PGE2 generation (**C**). Values displayed are mean ± S.E.M (*n* = 5), * *p* < 0.05 vs. control and # *p* < 0.05 vs. LPS.

**Figure 4 molecules-27-03784-f004:**
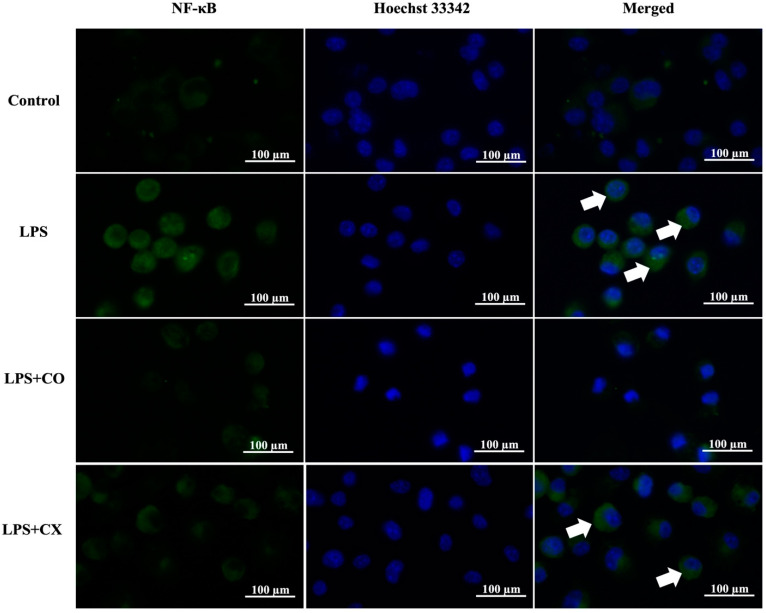
Effects of CO on NF-κB p65 nuclear activation. Cells were treated with 2% CO or 10 µM CX in the presence or absence of LPS for 24 h. Cells were stained with NF-κB p65 antibodies (green) and Hoechst 33342 (blue). Nuclear activation of NF-κB p65 (white arrow) was assessed using a fluorescent microscope.

**Figure 5 molecules-27-03784-f005:**
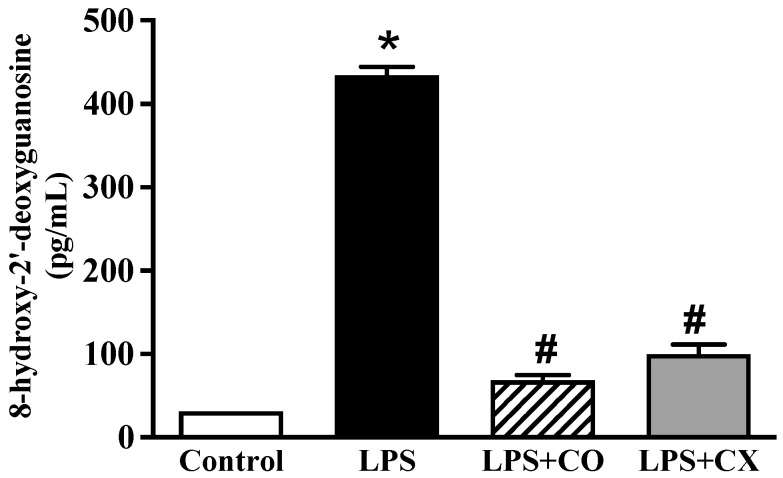
Effects of CO on DNA damage. Cells were treated with 2% CO or 10 µM CX with or without LPS. After 24 h, concentration of 8-OHdG was evaluated using an ELISA kit. Values displayed are mean ± S.E.M (*n* = 5), * *p* < 0.05 vs. control and # *p* < 0.05 vs. LPS.

**Figure 6 molecules-27-03784-f006:**
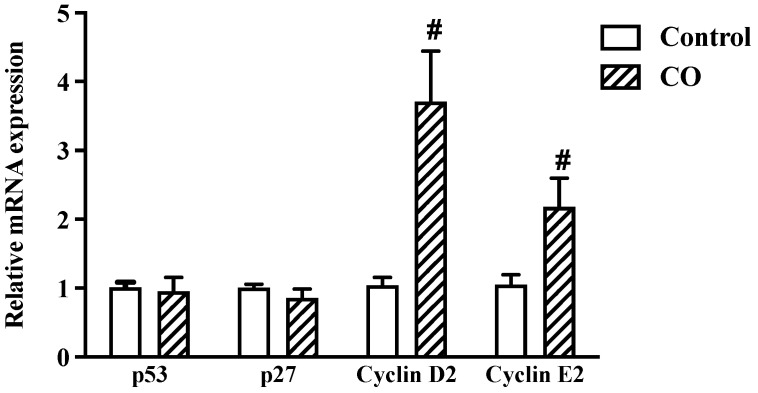
Effects of CO on cell-cycle regulators. Cells were treated with 2% CO or 10 µM CX with and without LPS for 24 h. mRNA expression of cell-cycle regulators was examined by real-time PCR. Values displayed are mean ± S.E.M (*n* = 5), # *p* < 0.05 vs. LPS.

**Figure 7 molecules-27-03784-f007:**
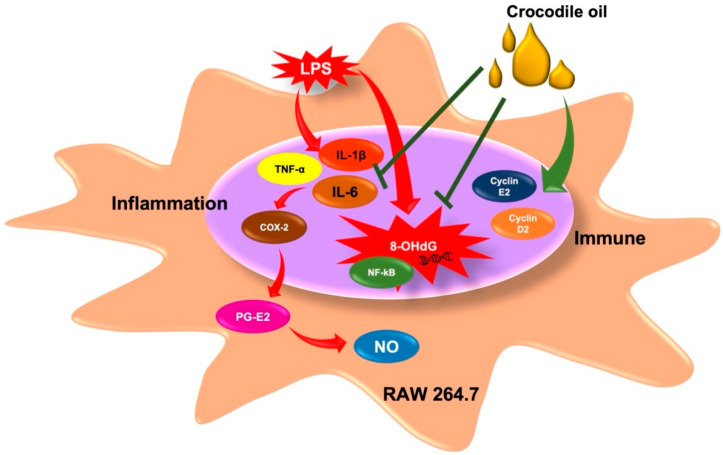
Inhibitory effects of CO on inflammatory status and its mechanisms.

**Table 1 molecules-27-03784-t001:** Fatty acid composition of crocodile oil.

Fatty Acid	Amount in Crocodile Oil (g/100 g)
Butyric acid (C4:0)	ND
Caproic acid (C6:0)	ND
Caprylic acid (C8:0)	ND
Capric acid (C10:0)	ND
Undecanoic acid (C11:0)	ND
Laurie acid (C12:0)	0.05 ± 0.03
Tridecanoic acid (C13:0)	ND
Myristic acid (C14:0)	0.45 ± 0.00
Pentadecanoic acid (C15:0)	0.08 ± 0.00
Palmitic acid (C16:0)	21.98 ± 1.30
Heptadecanoic acid (C17:0)	0.13 ± 0.01
Stearic acid (C18:0)	5.35 ± 0.36
Arachidic acid (C20:0)	0.11 ± 0.00
Heneicosanoic acid (C21:0)	0.02 ± 0.01
Behenic acid (C22:0)	0.05 ± 0.03
Tricosanoic acid (C23:0)	0.95 ± 0.02
Lignoceric acid (C24:0)	0.02 ± 0.00
Saturated fat	28.63 ± 0.14
Myristoleic acid (C14:1)	0.10 ± 0.00
*cis*-10-Pentadecenoic acid (C15:1n10)	ND
Palmitoleic acid (C16:1n7)	4.43 ± 0.46
*cis*-10-Heptadecenoic acid (C17:1n10)	0.06 ± 0.00
*trans*-9-Elaidic acid (C18:1n9t)	0.15 ± 0.05
*cis*-9-Oleic acid (C18:1n9c)	40.87 ± 2.81
*cis*-11-Eicosenoic acid (C20:1n 11)	0.41 ± 0.01
Erucic acid (C22:1n9)	0.03 ± 0.00
Nervonic acid (C24:1n9)	0.04 ± 0.01
Monounsaturated fatty acid	46.02 ± 3.10
*trans*-Linolelaidic acid (C18:2n6t)	ND
*cis*-9,12-Linoleic acid (C18:2n6)	23.29 ± 2.70
gamma-Linolenic acid (C18:3n6)	0.17 ± 0.11
alpha-Linolenic acid (C18:3n3)	1.05 ± 0.27
*cis*-11,14-Eicosadienoic acid (C20:2)	0.21 ± 0.07
*cis*-8,11,14-Eicosatrienoic acid (C20:3n6)	0.25 ± 0.10
*cis*-11,14,17-Eicosatrienoic acid (C20:3n3)	0.24 ± 0.37
Arachidonic acid (C20:4n6)	ND
*cis*-13,16-Docosadienoic acid (C22:2)	0.03 ± 0.01
*cis*-5,8,11,14,17-Eicosapentaenoic acid (C20:5n3)	0.04 ± 0.01
4,7,10,13,16,19-Docosahexaenoic acid (C22:6n3)	0.17 ± 0.01
Polyunsaturated fatty acid	25.35 ± 3.01
Unsaturated fat	71.37 ± 0.14
*Trans* fat	0.15 ± 0.05
Omega-3	1.4 ± 0.00
Omega-6	23.70 ± 2.90
Omaga-9	40.93 ± 2.78

Values shown are mean ± S.E.M. (*n* = 3). ND = not detected.

## Data Availability

The datasets analyzed during the current study are available from the corresponding author upon reasonable request.

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
