# Peer review of "Crocodile Oil Modulates Inflammation and Immune Responses in LPS-Stimulated RAW 264.7 Macrophages"

_molecules, 2022, doi:10.3390/molecules27123784_

Round 1

Reviewer 1 Report

This manuscript identified Crocodile oil contains a high concentration of omega-9 fatty acids and a low concentration of saturated fatty acids. Inflammatory mediators are reduced, and DNA damage is improved by reducing IL-6 production and gene expression and by lowering inflammatory mediators. This has meaningful implications for the utilization and product development, crocodile oil might thus be produced as a nutraceutical supplement to help avoid inflammatory diseases. This manuscript is well structured and informative, while for MS itself, there are some issues, which are listed below.

  1. Line 19, “as well as in wound healing”, delete “in”.
  2. Line 47, it appears that your sentence or clause uses an incorrect form of the verb include. Consider changing it into “including“.
  3. Line 79, “in the control group”
  4. Line 81, IL-6 production was considerably reduced, how much reduced exactly?
  5. Line 90, “with or without LPS “
  6. Line 128,“ctivation“ seems to be misspelled, it should be “activation”.
  7. Line 142, “with CO-treated cells “should be changed to “in CO-treated cells “
  8. Line 191, “for example, “
  9. I hope you can add some outlook on the application side of crocodile oil.
  10. The introduction of crocodile oil in the first paragraph of the introduction section is a bit simple, and should be enriched with the introduction of content about crocodile oil, such as the results of previous related studies.
  11. The introduction on the dangers of inflammation should also be briefly mentioned.
  12. Anti-inflammatory and anti-immune studies of other related oils should also be briefly mentioned in the introductory section.
  13. In the method section about the extraction of crocodile oil is not written in enough detail, I am not very clear about how exactly crocodile oil is extracted.
  14. Line 240,the MTT assay should be cited in the relevant references.
  15. Line 253,what concentration of lipopolysaccharide was used to stimulate the cells? The authors should have written clearly.
  16. Line 281,I think this part of the method should be more detailed, for example, how to extract the RNA from the cells and what is the procedure of quantitative real-time PCR?

Author Response

Response to Referee’s Comments

We greatly appreciate the reviewer’s comment which have indeed helped to improve the quality of this manuscript. The detail is present as suggested in a newly revised manuscript and the changes have made as highlighted in “red font”. We hope our revision has improved the paper to a satisfactory level.

Response to Referee’s Comments

We greatly appreciate the reviewer’s comment which have indeed helped to improve the quality of this manuscript. The detail is present as suggested in a newly revised manuscript and the changes have made as highlighted in “red font”. We hope our revision has improved the paper to a satisfactory level.

Reviewer #1

Major comments:

  1. Line 19, “as well as in wound healing”, delete “in”

As suggested, we have deleted “in” from the sentence in “Abstract” section page 1, line 19.

  1. Line 47, it appears that your sentence or clause uses an incorrect form of the verb include. Consider changing it into “including”.

We have deleted the phase “include carbon dioxide” and the sentence changed from “Other cosmetic goods, including oils, pain relievers (such as pain relief balm), and skin lotion, include carbon dioxide.” to “Other cosmetic goods, including oils, pain relievers (such as pain relief balm), and skin lotion.” in “Introduction” section page 2, line 48-49.

  1. Line 79, “in the control group”

As suggested, we have add the word “the” into the sentence “The results show that IL-6, IL-1b, and TNF-a levels were higher in LPS group than in the control group (Figure 1A–C), indicating that LPS promote inflammation.” in “Results” section page 4, line 87.

  1. Line 81, IL-6 production was considerably reduced, how much reduced exactly?

Figure 1A showed CO reduced IL-6 production by CO at 0.25-2% for 1.4-3.6 times compared with the LPS treated cells. Thus, we have changed the sentence from “Furthermore, IL-6 production was considerably reduced by CO at 0.25-2% compared to cells stimulated by LPS (Figure 1A).” to “Furthermore, IL-6 production was considerably reduced by CO at 0.25-2% for 1.4-3.6 times compared to cells stimulated by LPS (Figure 1A).” in “Results” section page 4, line 89-90.

  1. Line 90, “with or without LPS”

As suggested, we have deleted “of” from the sentence in “Results” section page 4, line 98.

  1. Line 128,“ctivation“ seems to be misspelled, it should be “activation”.

As suggested, we have added “a” into the word activation in “Results” section page 6, line 139.

  1. Line 142, “with CO-treated cells “should be changed to “in CO-treated cells”

As suggested, we have changed the pharse “with CO-treated cells” to “in CO-treated cells” in “Results” section page 7, line 153.

  1. Line 191, “for example,”

We have add comma(,) after the word for example in “Discussion” section page 8, line 203.

  1. I hope you can add some outlook on the application side of crocodile oil.

            As suggested, we have re-written  introduction part of crocodile oil in “Introduction” section page 1-2, line 45-50. We also added these 3 additional references in “References” section page 12, line 349-355.

  1. Li, H.L.; Chen, L.P.; Hu, Y.H.; Qin, Y.; Liang, G.; Xiong, Y.X.; Chen, Q.X. Crocodile oil enhances cutaneous burn wound healing and reduces scar formation in rats. Acad Emerg Med 2012, 19, 265-273, doi:10.1111/j.1553-2712.2012.01300.x.
  2. Tang, L.; Qin, M. The medicinal research and development prospects of crocodile. World Health Digest (in Chinese) 2007, 4, 66-68.
  3. Keapai, W.; Apichai, S.; Amornlerdpison, D.; Lailerd, N. Evaluation of fish oil-rich in MUFAs for anti-diabetic and anti-inflammation potential in experimental type 2 diabetic rats. Korean J Physiol Pharmacol 2016, 20, 581-593, doi:10.4196/kjpp.2016.20.6.581.

  1. The introduction of crocodile oil in the first paragraph of the introduction section is a bit simple, and should be enriched with the introduction of content about crocodile oil, such as the results of previous related studies.

            As suggested, we have re-written  introduction part of crocodile oil in “Introduction” section page 1-2, line 45-50. We also added these 3 additional references in “References” section page 12, line 349-355.

  1. Li, H.L.; Chen, L.P.; Hu, Y.H.; Qin, Y.; Liang, G.; Xiong, Y.X.; Chen, Q.X. Crocodile oil enhances cutaneous burn wound healing and reduces scar formation in rats. Acad Emerg Med 2012, 19, 265-273, doi:10.1111/j.1553-2712.2012.01300.x.
  2. Tang, L.; Qin, M. The medicinal research and development prospects of crocodile. World Health Digest (in Chinese) 2007, 4, 66-68.
  3. Keapai, W.; Apichai, S.; Amornlerdpison, D.; Lailerd, N. Evaluation of fish oil-rich in MUFAs for anti-diabetic and anti-inflammation potential in experimental type 2 diabetic rats. Korean J Physiol Pharmacol 2016, 20, 581-593, doi:10.4196/kjpp.2016.20.6.581.

  1. The introduction on the dangers of inflammation should also be briefly mentioned.

We have mentioned about inflammatory effectes in “Introduction” section page 7, line 191.

  1. Anti-inflammatory and anti-immune studies of other related oils should also be briefly mentioned in the introductory section.

We have mentioned about anti-inflammatory and anti-immune studies of other related oil in “Introduction” section page 2, line 53-55.

  1. In the method section about the extraction of crocodile oil is not written in enough detail, I am not very clear about how exactly crocodile oil is extracted.

We have re-writed the process of making CO in “Materials and Methods” section in page 9, line 234-247.

  1. Line 240,the MTT assay should be cited in the relevant references.

We have added the MTT assay relevant reference in “Materials and Methods” section page 10, line 257.

  1. Ontawong, A.; Duangjai, A.; Srimaroeng, C. Coffea arabica bean extract inhibits glucose transport and disaccharidase activity in Caco-2 cells. Biomed Rep 2021, 15, 73, doi:10.3892/br.2021.1449.

  1. Line 253,what concentration of lipopolysaccharide was used to stimulate the cells? The authors should have written clearly.

We have added the concentration of LPS  in “Materials and Methods” section page 10, line 271.

  1. Line 281,I think this part of the method should be more detailed, for example, how to extract the RNA from the cells and what is the procedure of quantitative real-time PCR?

As suggest, we have add briefly RNA and cDNA extraction method “Total RNA was extracted and purified from RAW264.7 cells using TRIzol reagent (Thermo Fisher Scientific, Waltham, MA, USA). First-strand cDNA was obtained using a SensiFAST™ cDNA synthesis kit (Bioline, London, UK).” in “Materials and Methods” section page 11, line 300-302.

Reviewer 2 Report

This manuscript describes the potential benefits of crocodile oil against inflammation. Overall the methodologies are appropriate and conclusion is supported by the results. However, extensive english editing on sentence structure and organization is needed to make it readable, especially abstract.  

For all graphs, the concentration of celecoxib (CX) needed to be in molar concentration. 

What vehicle is used to dissolve celecoxib?

What vehicle is used to dissolve crocodile oil? Does the concentration of vehicle used have any anti-inflammatory response?

If no vehicle is used, is 2% crocodile oil miscible in cell culture medium?

For results, it is better to start with " To study XXXX,  we performed XXX. Our results show that XXXX"

What passage number of RAW cells are being used?

Please change CO(percentage) to CO(%). Not sure whether it is w/w or v/v

Figure 1C. Based on the error bars, 2% CO should be significant?

I assume Nitric oxide is measured by Griess reagent?

Did the authors measured iNOS mRNA and protein expression since it affects NO production.

Figure 3B, it will be good to show somewhere it is COX-2 mRNA expression. 

Did you measure the protein expression of COX-2?

Figure 4 . Scale bar is needed. Please put arrows on which cells has p65 nuclear translocation.

Did you do subcellular fractionation followed by western blot to support your claims?

Did you test any NF-kB inhibitors?

It will be good to have a schematic diagram on the mechanism of action of crocodile oil based on your study

Author Response

Response to Referee’s Comments

We greatly appreciate the reviewer’s comment which have indeed helped to improve the quality of this manuscript. The detail is present as suggested in a newly revised manuscript and the changes have made as highlighted in “red font”. We hope our revision has improved the paper to a satisfactory level.

Reviewer #2

Major comments:

  1. For all graphs, the concentration of celecoxib (CX) needed to be in molar concentration. 

We agree with the reviewer that we should change the concentration of celecoxib to molar concentration. Thus, we have rewritten as suggested in “Figure 1.

  1. What vehicle is used to dissolve celecoxib?

            In this study, we used 100% DMSO to dissolve the stock of celecoxib at 200 mg/ml or 524 mM.

  1. What vehicle is used to dissolve crocodile oil? Does the concentration of vehicle used have any anti-inflammatory response?

            In this study we used 0.5% propylene glycol to dissolved crocodile oil and propylene glycol at this dose have no anti-inflammatory effect.

  1. If no vehicle is used, is 2% crocodile oil miscible in cell culture medium?

In this study we used 0.5% propylene glycol to dissolved crocodile oil.

  1. For results, it is better to start with " To study XXXX,  we performed XXX. Our results show that XXXX"

We agree with the reviewer that result section should be re-writed. Thus, we have rewritten as suggested in “Result” section.

  1. What passage number of RAW cells are being used?

            Cells in passage 2nd-22nd were used in this study.

  1. Please change CO(percentage) to CO(%). Not sure whether it is w/w or v/v

We agree with the reviewer that we should change CO(percentage) to CO(%). Thus, we have rewritten as suggested in “Result” section.

  1. Figure 1C. Based on the error bars, 2% CO should be significant?

            We have re-checked the statistically significant but 2% CO did not significantly differ between the LPS group.

  1. I assume Nitric oxide is measured by Griess reagent?

            This study used Griess reagent to determine nitric oxide levels.

  1. Did the authors measured iNOS mRNA and protein expression since it affects NO production.

            This study  did not determine iNOS mRNA expression because we use Griess reagent to determine nitric oxide levels.

  1. Figure 3B, it will be good to show somewhere it is COX-2 mRNA expression. 

            We have added COX-2 in y-axis of figure 3B.

  1. Did you measure the protein expression of COX-2?

In this study we did not determine COX-2 protein expression.

  1. Figure 4. Scale bar is needed. Please put arrows on which cells has p65 nuclear translocation.

            As suggested, we have added scale bar and put arrows in figure 4.

  1. Did you do subcellular fractionation followed by western blot to support your claims?

            In this study we did not performed western blot analysis to determine NFkB protein expression because we have show the fluorescence NFkB p65 staining.

  1. Did you test any NF-kB inhibitors?

            In this study, we did not test NF-kB inhibitors. However, a previous study reported that celecoxib suppressed NF-κB-regulated COX-2 and cyclin D1 protein expression. Therefore, we used celecoxib as a positive control. 

  1. It will be good to have a schematic diagram on the mechanism of action of crocodile oil based on your study

We have added graphical abstract in “Discission” section in page 8, line 175.

Reviewer 3 Report

the manuscript seems to be very interesting and in line with the topics of the journal.

I think that some points need to be revised before publication:

the title is not correct for the quality of the paper. please modify and highlight it to fully address the mechanism of action investigated.

the chemical composition of the CO must be improved by adding NMR analysis or HPLC-MS, UHPLC-HRMS or LC-MS, HPLC-DAD analysis. the quality of bio data must include a more detailed chemical composition.

Author Response

Response to Referee’s Comments

We greatly appreciate the reviewer’s comment which have indeed helped to improve the quality of this manuscript. The detail is present as suggested in a newly revised manuscript and the changes have made as highlighted in “red font”. We hope our revision has improved the paper to a satisfactory level

Reviewer #3

Major comments:

  1. the title is not correct for the quality of the paper. please modify and highlight it to fully address the mechanism of action investigated.

            As suggested, we have changed the title from “Crocodile Oil Modulations an Inflammation and Immune Response in LPS-stimulated RAW 264.7 Macrophages” to “Crocodile Oil Modulates an Inflammation and Immune Response in LPS-stimulated RAW 264.7 Macrophages”.

  1. The chemical composition of the CO must be improved by adding NMR analysis or HPLC-MS, UHPLC-HRMS or LC-MS, HPLC-DAD analysis. the quality of bio data must include a more detailed chemical composition.

We have re-writed the process of making CO in “Materials and Methods” section in page 9, line 235-252 and also added the fatty acid composition of crocodile oil information in “Results” section in Table 1 in page 2-4.

Round 2

Reviewer 2 Report

For Figure 4, how to see NF-kB activation? I am assuming NF-kB nuclear translocation, however I cannot see any obvious nuclear translocation of p65. The only thing I can see is p65 expression

Author Response

Response to Referee’s Comments

We greatly appreciate the reviewer’s comment which have indeed helped to improve the quality of this manuscript. The detail is present as suggested in a newly revised manuscript. We hope our revision has improved the paper to a satisfactory level.

Reviewer #2

Major comments:

  1. For Figure 4, how to see NF-kB activation? I am assuming NF-kB nuclear translocation, however I cannot see any obvious nuclear translocation of p65. The only thing I can see is p65 expression.

A previous study reported that increasing p65 NF-κB expression was correlated to NF-κB activation*. Thus, in this study, we used p65 NF-κB primary antibody to stain the p65 subunit of NF-κB dimer to imply the NF-κB activation. Moreover, we have rearranged Figure 4 as suggested.

* Weichert W, Boehm M, Gekeler V, Bahra M, Langrehr J, Neuhaus P, Denkert C, Imre G, Weller C, Hofmann HP, Niesporek S, Jacob J, Dietel M, Scheidereit C, Kristiansen G. High expression of RelA/p65 is associated with activation of nuclear factor-kappaB-dependent signaling in pancreatic cancer and marks a patient population with poor prognosis. Br J Cancer. 2007 Aug 20;97(4):523-30. doi: 10.1038/sj.bjc.6603878.